# Postprandial Glycaemia, Insulinemia, and Lipidemia after 12 Weeks' Cheese Consumption: An Exploratory Randomized Controlled Human Sub-Study

Louise Kjølbæk [1,*], Farinaz Raziani [2], Tine Tholstrup [1], Rosa Caroline Jullie Rudnicki [3], Christian Ritz [4], Arne Astrup [5] and Anne Raben [1,6]

1   Department of Nutrition, Exercise and Sports, Faculty of Science, University of Copenhagen, Rolighedsvej 26, 1958 Frederiksberg C, Denmark
2   Y-mAbs Therapeutics, Agern Alle 11, 2970 Hørsholm, Denmark
3   Center for Rheumatology and Spine Diseases, Copenhagen University Hospital Rigshospitalet, Juliane Maries vej 10, 2100 Copenhagen Ø, Denmark
4   National Institute of Public Health, University of Southern Denmark, Studiestræde 6, 1455 Copenhagen K, Denmark
5   Healthy Weight Center, Novo Nordisk Foundation, Tuborg Havnevej 19, 2900 Hellerup, Denmark
6   Clinical Research, Copenhagen University Hospital-Steno Diabetes Center Copenhagen, Borgmester Ib Juuls Vej 83, 2730 Herlev, Denmark
*   Correspondence: louisekjoelbaek@nexs.ku.dk; Tel.: +45-35-33-14-62

**Abstract:** Some populations are recommended to consume low-fat dairy, although the evidence behind replacing high-fat with low-fat dairy products is limited. This exploratory sub-study investigated the effect of cheese with different fat content on postprandial changes in type-2-diabetes risk markers. Following 12-week cheese or jam intake, a 4 h meal test was conducted with 37 participants. Test meals included bread and either: 80 g regular-fat cheese (REG), 80 g reduced-fat cheese (RED) or 25 g jam (CHO). Postprandial blood was drawn and appetite sensations registered. Time-meal interactions were not observed for glucose and insulin, but for triglycerides (TG) and free fatty acids (FFA). Pairwise comparisons showed $0.17 \pm 0.07$ mmol/L ($p = 0.044$) and $0.25 \pm 0.07$ mmol/L ($p = 0.002$) higher TG at 180 and 240 min, respectively, and $94 \pm 37$ mmol/L ($p = 0.029$) higher FFA at 180 min for REG compared with RED. Compared with CHO, intake of both cheese meals reduced insulin and glucose (main effects of meal, both $p \leq 0.011$) and increased FFA and TG at certain time points. In conclusion, intake of cheese with a regular, compared with reduced, fat content did not affect glucose, insulin and appetite, but increased TG and FFA.

**Keywords:** postprandial; acute; insulin; glucose; triglycerides; free fatty acids; cheese; diabetes; appetite; dairy

## 1. Introduction

Type-2-diabetes (T2D) is a metabolic disease which has reached epidemic proportions. The prevalence of T2D for all age groups has increased worldwide, primarily due to the rise in obesity [1]. Dietary modifications, physical activity, and weight loss are the main treatment options for individuals with T2D, although the best nutritional advice for weight loss is still under debate [2]. However, an effective treatment of hyperglycaemia for persons diagnosed with T2D, even without a weight loss, is to replace carbohydrates [3], for example with fat-sources like dairy products. Several meta-analyses about dairy intake and T2D have been conducted. A meta-analysis of 22 prospective cohort studies (579,832 individuals) found that a higher total dairy intake was linearly associated with lower T2D risk [4]. The beneficial health effect may be attributable to the high content of calcium, magnesium, vitamin D, and whey proteins found within the dairy food matrix [5]. For fermented or ripened dairy products, results from both human and animal studies suggest that the

presence of branch-chained amino acids is involved in the regulation of insulinemia [6]. However, dairy products are a heterogeneous group where, for example, cheese contains a high amount of saturated fat. For many years, a high content of both total and saturated fat has been hypothesised to increase risk of cardiovascular diseases (CVD), although the evidence seems uncertain [3].

For cheese intake in particular, the former mentioned meta-analysis of cohorts [4] did not observe any association with T2D. Human intervention studies investigating the effect of cheese intake on T2D risk markers have shown mixed results. In healthy participants, one study observed that fasting glucose concentration was higher and fasting insulin concentration unaffected after cheese consumption, compared to butter [7]. Another study did not observe any effect of cheese intake on fasting glucose concentration, but instead, a reduction in fasting insulin concentration, compared to a non-dairy control diet [8]. In mildly hypercholesterolaemic participants, one study found no change in fasting glucose or insulin concentrations after cheese intake, compared to butter [9]. When interpreting these results, it may be relevant to focus on the substitution of one product with another product. Recently, Ibsen et al. [10] observed from a cohort study that substitution of low-fat milk by whole-fat milk was not associated with risk of T2D, but surprisingly they found that substitution of low-fat yoghurt by whole-fat yoghurt was associated with a lower risk of T2D. Unfortunately, substitution between whole-fat (regular-fat) and low-fat (reduced-fat) cheeses was not included in their analysis, but it was speculated whether a synergistic effect between fat and fermentation could explain the findings. Earlier, we reported that a 12-week intake of cheese with different fat content did not affect fasting insulin, glucose and homeostatic model assessment for insulin resistance (HOMA-IR) [11].

Although consumption of milk and dairy products may protect against the most prevalent non-communicable diseases [12], dietary guidelines (e.g., the Nordic Nutrition Recommendation 2012 and the Dietary Guidelines for Americans 2020–2025) specifically recommend intake of low-fat dairy products to reduce risk of CVD and T2D. However, the current evidence does not support this recommendation and an expert panel suggested in 2017 that future research should focus on whole foods rather than nutrients because the food matrix may have the potential to influence the health effects [6]. The current sub-study investigates the acute effects of real-life meals reflecting whole food substitution strategies. Such meals cannot be energy- and nutrient-matched, so they may affect postprandial appetite and substrate metabolism differently. We and others have investigated effects of the dairy matrix on the postprandial response [13–15]. However, unravelling the specific structure or texture of the matrix per see is complicated and another question is if the matrix matters more than the fat content. None of the postprandial matrix studies compared cheese with similar matrix, but different fat content. Thus, the effect of cheese fat content on postprandial indicators of T2D risk markers requires further exploration.

The aim of this exploratory sub-study was to compare the effect of meals containing cheese with different fat content or carbohydrate-rich foods on 4 h postprandial glucose, insulin, triglyceride (TG) and free fatty acids (FFA) concentrations as well as subjective appetite sensations, after 12 weeks habituation to the diets.

## 2. Materials and Methods

### 2.1. Study Design

This exploratory sub-study was carried out in a sub-group of participants participating in a 12-week parallel randomized controlled trial conducted from February 2014 to May 2015 (described elsewhere [11]). The main intervention study was conducted during four periods and this explorative sub-study (a meal test) was added to the main study protocol in November 2014. Thus, participants included in this sub-study were recruited during the last period of the main intervention study, i.e., January to May 2015, and they gave their written informed consent after receiving written and oral information about the sub-study. This sub-study was conducted at Department of Nutrition, Exercise and Sports, University of Copenhagen, Frederiksberg, Denmark. The study was conducted according

to the guidelines of the Declaration of Helsinki, approved by the Ethical Committee, Region H, Denmark (H-4-2013-099) and registered in the Danish Data Protection Agency (2007-54-0269) and at clinicaltrials.gov (NCT02616471).

This sub-study included one test day in which participants took part in an acute 4 h meal test. The test day was scheduled on the last day of the participants' 12-week intervention period. Conditions before and on the sampling days were standardized. Thus, participants were requested to be fasting for 12 h (a maximum of 0.5 L water intake was allowed), not to perform extreme physical activity for 24 h, and not to drink alcohol for 24 h prior to the test day. Participants arrived at the Department at approximately 07:30 in the morning. Upon arrival the participants had a peripheral venous catheter inserted in an antecubital vein for postprandial blood sampling, and after 10 min rest in a supine position, a fasting blood sample (time 0 min) was drawn. The test meal was served as a breakfast and participants were asked to ingest the whole meal within 15 min. In addition, participants were instructed to eat at a constant pace and to distribute the meal over the period of 15 min. In the following 4 h blood samples were withdrawn at 30, 60, 90, 120, 180, and 240 min after breakfast commenced. Participants remained resting throughout the study and were not allowed to consume any foods until after the last blood sample was taken. However, they were given 100 mL of water after 120 min, which they had to consume. Participants were also asked to fill out visual analogue scale (VAS) at each time point for assessment of subjective appetite sensations [16]. Further, palatability of the test meals was assessed by VAS. The primary outcome of this exploratory sub-study was risk markers of T2D, i.e., glucose and insulin. Secondary outcomes were TG and FFA. Explorative outcomes were subjective appetite sensations.

### 2.2. Participants

In brief, participants were recruited through advertisements in newspapers in the Copenhagen area. They were included in the main intervention study if they fulfilled the following criteria: 18–70 years of age, body mass index (BMI) of 18.5–37.5 kg/m$^2$, waist circumference > 80 cm for women and >94 cm for men, and additionally ≥1 risk factor for the metabolic syndrome as described elsewhere [11]. These criteria were assessed at screening (before the 12-week intervention period was initiated) and exclusion criteria are described elsewhere [11]. At screening, all included participants were randomly allocated to one of three intervention groups: regular-fat cheese (REG), reduced-fat cheese (RED), or a non-cheese carbohydrate-rich (CHO) diet for 12 weeks. Allocation ratio of the main study was 1:1:1 with randomization sequences stratified by sex and smoking status. After the 12-week intervention period, participants in this exploratory sub-study were served a test meal corresponding to the diet he or she had been consuming during the 12-week intervention.

### 2.3. Test Meals

The test meals were served as breakfast meals. The participants in the REG group were served 40 g full-fat Danbo (Riberhus, 25% fat, Arla Foods, Viby J, Denmark) and 40 g cheddar (Sharp Cheddar, 32% fat, Lactalis, Scotland) cheeses. The participants in the RED group were served 40 g reduced/low-fat Danbo (Riberhus, 13% fat, Arla, Denmark) and 40 g cheddar (Sharp Cheddar, 16% fat, Lactalis, Scotland) cheeses. In both groups, cheese was served with 75 g white wheat bread (Kohberg Bakery Group A/S, Bolderslev, Denmark). The CHO test meal was iso-energetic to the REG test meal and consisted of 165 g white wheat bread (Kohberg, Denmark) and 25 g of sugar-sweetened raspberry jam containing 50 g fruit/100 g and 45 g added sugar/100 g (Fynbo Foods A/s, Vrå, Denmark). All three test meals were served with 25 g cucumber and 250 mL of water. Energy content and macronutrient composition of the test meals were calculated using the software program Dankost (Dankost Pro, Copenhagen, Denmark). The composition of the test meals is presented in Table 1. It was not possible to blind the study participants due to

the nature of the meals. However, the lab technicians performing the blood analyses and researchers performing statistical analyses were blinded to the meal allocation.

**Table 1.** Composition of the three breakfast meals [1].

|  | REG | RED | CHO |
|---|---|---|---|
| Energy (kJ) | 2001 | 1787 | 2000 |
| Meal weight (g) | 430 | 430 | 465 |
| Energy density (kJ/g) | 4.7 | 4.2 | 4.3 |
| Fat |  |  |  |
| (E%) | 46.4 | 33.7 | 6.5 |
| (g) | 25.1 | 16.3 | 3.5 |
| Saturated fat (g) | 14.6 | 4.6 | 0.4 |
| Carbohydrate |  |  |  |
| (E%) | 33.0 | 37.0 | 83.4 |
| (g) | 38.1 | 38.0 | 96.0 |
| Dietary fibre (g) | 1.8 | 1.8 | 1.8 |
| Protein |  |  |  |
| (E%) | 20.6 | 29.3 | 10.1 |
| (g) | 24.2 | 30.8 | 11.9 |

[1] Nutrition information provided by the manufacturers (Arla, Lactalis, Kohberg, and Fynbo). CHO, carbohydrate-rich meal; RED, reduced-fat cheese; REG, regular-fat cheese; E%, percentage of energy.

*2.4. Blood Samples*

Blood samples were drawn for analyses of serum insulin, plasma glucose, serum TG, and serum FFA. Insulin was measured by chemiluminescent immunoassay with an Immulite 1000 (Siemens Medical Solution Diagnostics). Interassay and intraassay CVs for insulin were 3.5% and 4.2%, respectively. Glucose concentration was measured with an enzymatic procedure (ABX Pentra Glucose HK CP) and analysed with an ABX Pentra 400 Chemistry Analyzer (Horiba ABX). Interassay and intraassay CVs for glucose were 2.5% and 0.8%, respectively. TG concentration was assessed by an enzymatic procedure (glycerol-3-phosphate oxidase-phenol 1 aminophenazone). The analysis was carried out with an ABX Pentra 400 Chemistry Analyzer (Horiba ABX). Interassay and intraassay CVs for TG were 3.5% and 3.8%, respectively. FFA was quantitative determined, by an enzymatic colorimetric method using the NEFA-HR(2) assay from Wako Chemicals GmbHJ, Germany. The analysis was carried out with an ABX Pentra 400 from Horiba ABX, France. Interassay and intraassay CVs for FFA were 2.0% and 1.7%, respectively. Analyses of total cholesterol and high-density lipoprotein (HDL) cholesterol (only reported as descriptive fasting values) are described elsewhere [11].

*2.5. Appetite and Palatability Registration*

Subjective sensations were recorded using VAS before and during the 4 h meal test. An electronic VAS tablet (Lenovo Group Limited, Beijing, China) with the software Acqui (Acqui, DK) was used with questions to access their subjective feelings. For each question a mark should be made on a line (100 mm in length) with extremes anchored at each end expressing the most positive and the most negative rating of the participant's sensations. For subjective sensations the following parameters were assessed: appetite (satiety, hunger, fullness and prospective consumption), thirst, desire to eat meat/fish, desire for specific tastes (salty, fatty and sweet) and well-being. Palatability of the meal was also rated on a 100 mm VAS anchored with questions on smell, taste, off taste, look and overall

appearance of the meals. The use of VAS has previously been shown to be both reproducible and valid for measurement of appetite sensations [16,17].

*2.6. Statistical Analysis*

No power calculation was done for this explorative sub-study, as it was conducted as an add-on study to the main intervention study. All statistical analyses were performed using R version 4.0.5 (R Core Team, 2021) [18]. Descriptive data are presented as mean value with SD. Results are presented as mean value and SEM and *p*-values were considered significantly different at *p* < 0.05. For results presented as mean value with SD, see Supplementary Material (Table S1 and Figures S1–S3). Fasting values were tested for group differences. Continuous data were tested using one-way ANOVA or Kruskal–Wallis test and categorical data using Fishers exact test. All repeated outcomes were analysed as repeated measurements and as 4 h incremental area under the curve (iAUC) or 4 h incremental area over the curve (iAOC), i.e., the iAUC included all area below the curve and above fasting concentration (0 min). Area beneath fasting concentration was ignored. Repeated measurements were analysed through linear mixed models including a time-meal interaction, time and meal as fixed effects, participant as random effect and with adjustments for age, sex, BMI and fasting value. Serial correlation between time points within participant was modelled assuming a spatial Gaussian correlation structure. Results were reported as post hoc t-tests for comparisons of test meals when a main meal effect was observed or, if the time-meal interaction was significant, comparisons between test meals within each time point. For both main meal effects and time-meal interactions, *p*-value adjustment for pairwise comparisons was based on the single-step procedure proposed by Hothorn et al. [19] and implemented in the R extension package "multcomp". Non-repeated measurements (iAUC, iAOC and palatability rating) were analysed using linear mixed models including meal as fixed effect, participant as random effect and with adjustments for age, sex and BMI. Results were reported as model-based pairwise comparisons with *p*-value adjustment for pairwise comparisons as described above. For glucose, insulin, TG and FFA, secondary analyses excluding two participants with fasting glucose > 7.00 mmol/L were conducted. The palatability ratings of the three test meals differed for taste, so this parameter was included as covariate in the analyses of all appetite sensation variables. Due to the study design (being exploratory and not having sex included in the design, i.e., as a 2 × 3 factorial design) as well as the low number of women (5–9) and men (4–6) in each group, data were not analysed or reported disaggregated by sex.

## 3. Results

During the fourth recruitment period of the main study, 53 participants were included and 37 completed the meal test (Figure 1). At week 12, the completers (21 women and 16 men) had a mean BMI of 28.3 ± 4.1 kg/m$^2$ and a mean age of 53 ± 12 years. In addition to increased waist circumference, 12 completers did not have risk factors for metabolic syndrome when the meal test was conducted at week 12, and two women did not have a waist circumference > 80 cm although having other risk markers for the metabolic syndrome. Table 2 shows the characteristics of the participants in the three groups.

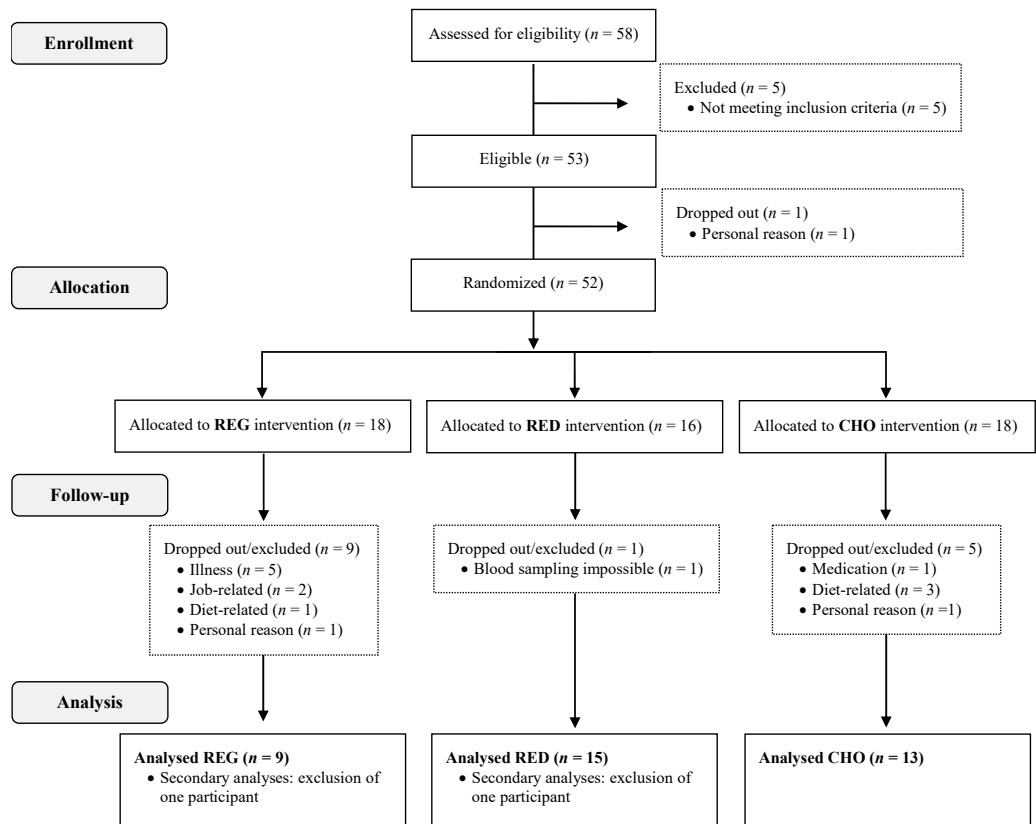

**Figure 1.** Flow chart. CHO, carbohydrate-rich meal; RED, reduced-fat cheese; REG, regular-fat cheese.

**Table 2.** Characteristics of the 37 participants who completed the test meal [1].

| | REG (*n* = 9) | RED (*n* = 15) | CHO (*n* = 13) | *p* [2] |
|---|---|---|---|---|
| Sex: women/men [n (%)] | 5 (56)/4 (44) | 9 (60)/6 (40) | 7 (54)/6 (46) | 1.0 |
| Age (years) | 51 ± 16 | 52 ± 12 | 57 ± 11 | 0.55 |
| Body mass index (kg/m$^2$) | 29.2 ± 4.0 | 27.8 ± 4.2 | 28.3 ± 4.2 | 0.73 |
| Smoking [n (%)] | 0 (0) | 2 (13) | 0 (0) | 0.33 |
| Waist circumference (cm) | 98.3 ± 11.1 | 97.8 ± 12.9 | 99.4 ± 13.3 | 0.95 |
| Systolic blood pressure (mmHg) | 127.7 ± 14.1 | 123.9 ± 17.6 | 124.3 ± 15.7 | 0.84 |
| Diastolic blood pressure (mmHg) | 81.4 ± 7.9 | 80.0 ± 9.1 | 78.9 ± 9.0 | 0.80 |
| Fasting triglycerides (mmol/L) | 1.47 ± 0.86 | 1.20 ± 0.49 | 0.99 ± 0.31 | 0.49 |
| Fasting total cholesterol (mmol/L) | 5.36 ± 1.59 | 4.99 ± 0.83 | 4.87 ± 0.69 | 0.45 |
| Fasting HDL cholesterol (mmol/L) | 1.53 ± 0.42 | 1.45 ± 0.28 | 1.51 ± 0.32 | 0.96 |
| Fasting glucose (mmol/L) | 6.02 ± 0.61 | 5.85 ± 0.90 | 5.61 ± 0.66 | 0.45 |
| Number of risk factors for MetS [3] 0/1/2/3/4 [n] | 1/2/4/2/0 | 5/5/2/2/1 | 6/2/4/1/0 | 0.50 |
| Fasting insulin (pmol/L) | 76.3 ± 37.3 | 61.8 ± 37.7 | 67.0 ± 45.9 | 0.50 |
| Fasting free fatty acids (μmol/L) | 499 ± 198 | 569 ± 120 | 485 ± 145 | 0.30 |

[1] Measurements from week 12 is presented as mean ± SD or count (percentage). [2] *p*-value for test of group difference using one-way ANOVA or Kruskal–Wallis test for continuous data and Fishers exact test for categorical data. [3] Number of risk factors for metabolic syndrome (systolic blood pressure ≥ 130 mmHg or diastolic blood pressure ≥ 85 mmHg, triglycerides ≥ 1.7 mmol/L, glucose ≥ 5.6 mmol/L, HDL cholesterol < 1.03 mmol/L and <1.29 mmol/L for men and women, respectively) in addition to increased waist circumference > 80 cm for women and >94 cm for men. CHO, carbohydrate-rich meal; HDL, high-density lipoprotein; MetS, metabolic syndrome; RED, reduced-fat cheese; REG, regular-fat cheese.

### 3.1. Insulin and Glucose

There were no significant time-meal interactions for postprandial glucose or insulin concentrations (Figure 2). However, main effects of meal on glucose ($p = 0.011$) and insulin ($p < 0.0001$) concentrations were observed. Pairwise comparisons showed that the postprandial glucose response was higher for the CHO meal compared with the REG ($0.62 \pm 0.22$ mmol/L, $p = 0.015$) and RED ($0.54 \pm 0.19$ mmol/L, $p = 0.013$) meals. Furthermore, an overall main meal effect was observed on glucose iAUC ($p = 0.014$). Pairwise comparisons showed a larger iAUC for the CHO meal compared with both cheese meals (REG: $p = 0.036$ and RED: $p = 0.025$). For insulin, pairwise comparisons showed that the postprandial response was higher for the CHO meal compared with the REG ($119.2 \pm 22.7$ pmol/L, $p < 0.0001$) and RED ($91.1 \pm 19.7$ pmol/L, $p < 0.0001$) meals. Additionally, an overall main meal effect was observed for insulin iAUC ($p = 0.009$), where pairwise comparisons showed a larger iAUC for the CHO meal compared with both cheese meals (REG: $p = 0.023$ and RED: $p = 0.019$).

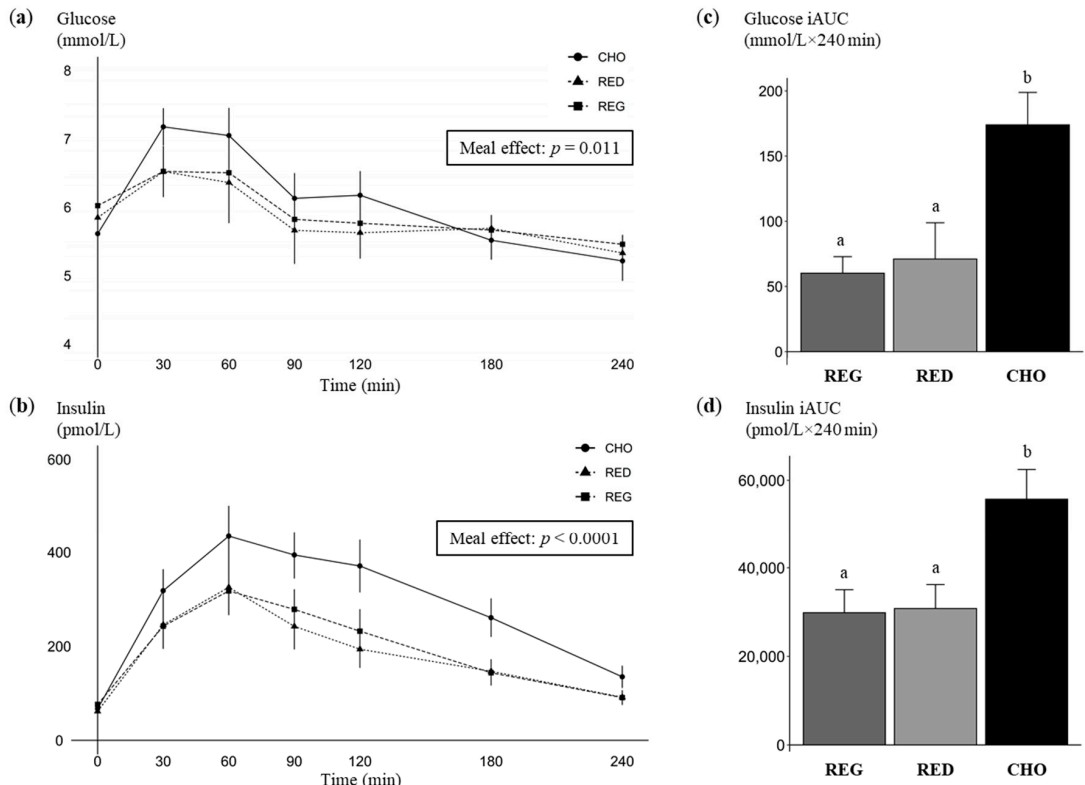

**Figure 2.** Postprandial blood responses of glucose and insulin after intake of regular-fat cheese (REG), reduced-fat cheese (RED) or carbohydrate-rich (CHO) meals ($n = 37$). (**a**,**b**) Unadjusted mean $\pm$ SEM of glucose (**a**) and insulin (**b**). Results were analysed by linear mixed model including a time-meal interaction, time and meal as fixed effects, participant as random effect and adjustments for age, sex, body mass index and fasting value. (**a**) Main meal effect was observed for glucose. Post hoc analyses with *p*-value adjustment for multiple comparisons showed higher glucose concentrations for CHO compared to RED ($p = 0.013$) and REG ($p = 0.015$). (**b**) Main meal effect was observed for insulin. Post hoc analyses with *p*-value adjustment for multiple comparisons showed higher insulin concentration for CHO compared to RED and REG (both $p < 0.0001$). (**c**,**d**) Unadjusted mean + SEM of glucose (**c**) and insulin (**d**). iAUCs were analysed by linear mixed model including meal as fixed effect, participant as random effect and adjustments for age, sex and body mass index. Overall main meal effects were observed (both $p \leq 0.014$) and post hoc analyses with *p*-value adjustment for multiple comparisons were conducted. Meals not sharing a common letter are significantly different ($p < 0.05$). iAUC, incremental area under the curve; RED, reduced-fat cheese; REG, regular-fat cheese; CHO, carbohydrate-rich meal.

The secondary analyses (excluding two participants with fasting glucose > 7.00 mmol/L) changed the results. After exclusion, time-meal interactions for glucose ($p$ = 0.0087) and insulin ($p$ = 0.0417) were observed. Post hoc comparisons showed differences between the CHO meal and RED or RED meals in the interval from 30 to 180 min, with no differences between REG and RED meals (data not shown). For iAUC, the exclusion did not change the overall results, but the differences between CHO and cheese meals became larger and more significant, i.e., smaller $p$-values (data not shown).

### 3.2. TG and FFA

The postprandial TG and FFA concentrations are shown in Figure 3. There were time-meal interactions for both TG and FFA (both $p$ < 0.0001). Pairwise comparisons showed at 180 and 240 min that REG meal resulted in higher TG concentration compared to RED (0.17 ± 0.07 mmol/L, $p$ = 0.044, and 0.25 ± 0.07 mmol/L, $p$ = 0.002, respectively) and CHO (0.39 ± 0.08 mmol/L, $p$ < 0.001, and 0.45 ± 0.08 mmol/L, $p$ < 0.001, respectively) meals. Furthermore, RED meal resulted in higher TG concentration compared to the CHO meal at both 180 and 240 min (0.22 ± 0.07 mmol/L, $p$ = 0.002, and 0.20 ± 0.07 mmol/L, $p$ = 0.006, respectively) (Figure 3a). For TG iAUC, an overall meal effect was observed ($p$ = 0.005). Pairwise comparisons showed a larger iAUC for REG compared to the CHO meal ($p$ = 0.004). For FFA, pairwise comparisons showed REG resulted in higher FFA concentrations compared to CHO at 120, 180 and 240 min (105 ± 38 mmol/L, $p$ = 0.015, 165 ± 38 mmol/L, $p$ < 0.001 and 247 ± 38 mmol/L, $p$ < 0.0001, respectively). Compared to RED, REG resulted in higher FFA concentrations at 180 min (94 ± 37 mmol/L, $p$ = 0.029). RED resulted in higher FFA concentrations compared to CHO at 240 min (178 ± 34 mmol/L, $p$ < 0.0001), however only a tendency was observed at 180 min ($p$ = 0.086). For FFA, the effect of meal on iAOC tended to differ ($p$ = 0.06) (Figure 3d).

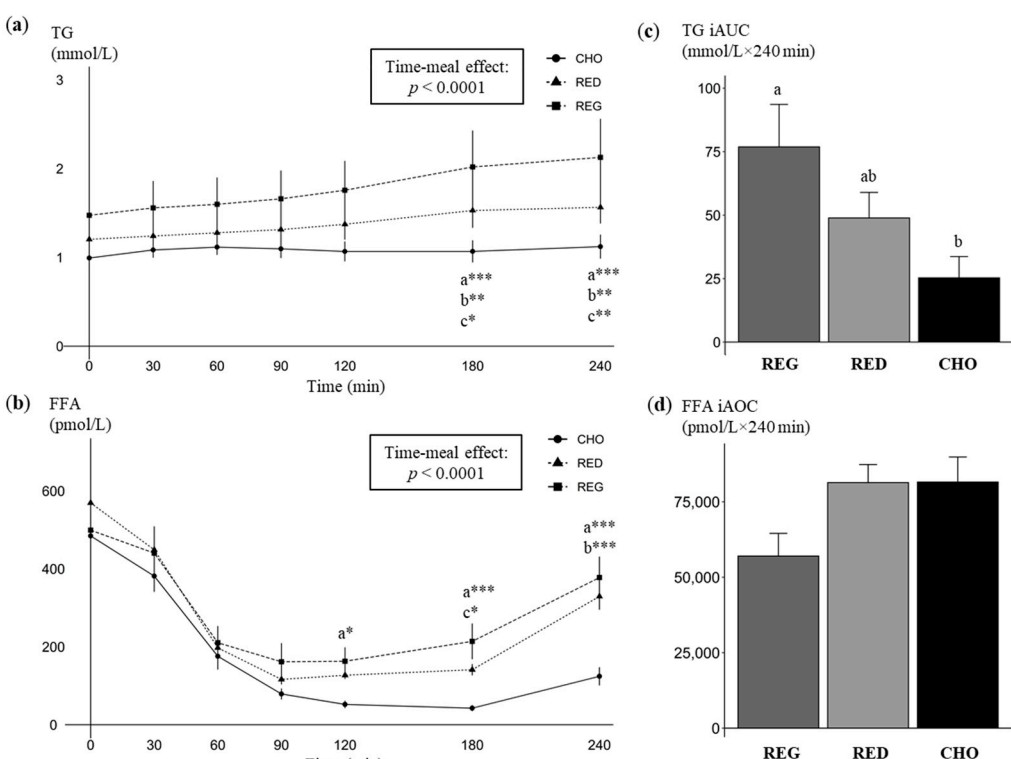

**Figure 3.** Postprandial blood responses of triglycerides (TG) and free fatty acids (FFA) after intake of regular-fat cheese (REG), reduced-fat cheese (RED) or carbohydrate-rich (CHO) meals (*n* = 37). (**a**,**b**) Un-adjusted mean ± SEM of TG (**a**) and FFA (**b**). Results were analysed by linear mixed model including a time-meal interaction, time and meal as fixed effects, participant as random effect and adjustments for age, sex, body mass index and fasting value. (**a**,**b**) Time-meal interactions were observed for TG and FFA.

Results of post hoc analyses with *p*-value adjustment for multiple comparisons are showed in the figure with a–c indicating significant differences at the given time point: a CHO compared with REG, b CHO compared with RED and c REG compared with RED at the following levels: * $p < 0.05$, ** $p < 0.01$, *** $p < 0.001$. (**c**,**d**) Unadjusted mean + SEM of TG (**c**) and FFA (**d**). iAUCs were analysed by linear mixed model including meal as fixed effect, participant as random effect and adjustments for age, sex and body mass index. Main meal effect was observed for TG ($p = 0.005$) and post hoc analyses with *p*-value adjustment for multiple comparisons were conducted. Meals not sharing a common letter are significantly different ($p < 0.05$). CHO, carbohydrate-rich meal; FFA, free fatty acids; iAOC, incremental area over the curve; iAUC, incremental area under the curve; RED, reduced-fat cheese; REG, regular-fat cheese; TG, triglycerides.

Exclusion of the two participants with fasting glucose > 7.00 mmol/L (secondary analyses) resulted in minor changes. For TG, the difference between REG and RED at 180 min disappeared ($p = 0.099$) and no change was observed for iAUC results (data not shown). For FFA, the difference between REG and RED disappeared ($p = 0.063$), as well as the tendency observed for iAOC (data not shown).

### 3.3. Palatability of the Test Meals

The three test meals were not rated differently in terms of odor, look, off taste and general appearance (Table 3). However, taste differed between two of the meals, and post hoc comparisons showed that the REG meal was rated less tasteful than the CHO meal ($28.5 \pm 9.5$ mm, $p = 0.014$).

**Table 3.** Palatability assessments of the three test meals [1,2].

| | REG (*n* = 9) | RED (*n* = 15) | CHO (*n* = 13) |
|---|---|---|---|
| Taste (mm) | $28.3 \pm 5.5$ | $37.9 \pm 5.7$ | $55.5 \pm 6.5$ * |
| Look (mm) | $63.6 \pm 9.3$ | $55.3 \pm 5.8$ | $67.7 \pm 6.8$ |
| Odor (mm) | $42.9 \pm 9.4$ | $50.5 \pm 4.9$ | $54.1 \pm 9.0$ |
| Off taste (mm) | $75.1 \pm 10.8$ | $55.8 \pm 11.2$ | $77.0 \pm 7.5$ |
| General appearance (mm) | $49.0 \pm 8.1$ | $52.8 \pm 5.8$ | $60.0 \pm 7.4$ |

[1] Mean $\pm$ SEM. [2] Ratings were evaluated by visual analogue scales from 0 to 100 mm. * Significantly different from REG ($p = 0.014$). CHO, carbohydrate-rich meal; RED, reduced-fat cheese; REG, regular-fat cheese.

### 3.4. Subjective Sensations of Appetite

Postprandial profiles of satiety and hunger are shown in Figure 4. There were neither time-meal interactions nor main effects of meal on any of the appetite sensation parameters (satiety, hunger, fullness and prospective consumption), thirst, desire to eat meat/fish, or desire for specific tastes like something salty, fatty or sweet (data not shown). For well-being, a time-meal interaction was observed ($p = 0.044$). Pairwise comparisons showed that well-being was $13 \pm 5$ mm lower at 240 min for RED compared with CHO ($p = 0.041$).

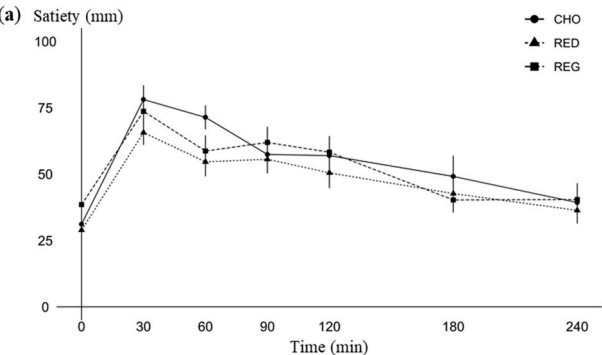

**Figure 4.** *Cont.*

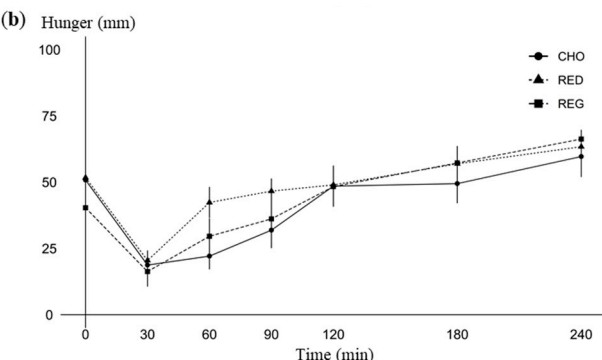

**Figure 4.** Postprandial appetite sensation of satiety (*n* = 37) and hunger (*n* = 36). Unadjusted mean ± SEM of satiety (**a**) and hunger (**b**) measured by visual analogue scale. Results were analysed by linear mixed model including a time-meal interaction, time and meal as fixed effects, participant as random effect and adjustments for age, sex, body mass index, fasting value and taste. Neither time-meal interactions nor main effect of meal were significant ($p > 0.05$). CHO, carbohydrate-rich meal; RED, reduced-fat cheese; REG, regular-fat cheese.

## 4. Discussion

The aim of the present study was to compare the 4 h postprandial changes in insulin, glucose, TG, FFA and subjective appetite sensations of cheese with different fat content or non-cheese carbohydrate-rich foods. Interestingly, REG and RED meals resulted in a similar glycemic and insulinemic response, whereas the two cheeses, as hypothesized, showed lower glucose and insulin response compared with the CHO meal. For TG, we observed not only that the CHO meal gave a lower TG concentration after 3 and 4 h compared to the cheese meals, but also that intake of RED resulted in a lower 3 and 4 h TG concentration than REG. For FFA, a difference between RED and REG was observed only at 3 h. Appetite sensations showed no difference between any of the three meals.

No highly similar postprandial studies with cheese varying in fat content have been conducted previously, but a few have compared dairy meals with carbohydrate-rich meals. For studies using real food, Thorning et al. [20] found higher 3 h glucose after a breakfast meal with cheese compared to a breakfast meal with carbohydrates. This was surprising and in contrast to our results. However, Thorning et al. only measured blood variables at one time point, 3 h after breakfast, and this methodological difference between the two studies is likely to explain the diverging results. Our finding was expected due to the higher amount of carbohydrates in the CHO meal (CHO, 96.0 g; RED, 38.0 g; REG, 38.1 g), but also because carbohydrates in this meal were primarily from white wheat bread, which generally is considered as having a high glycemic index (GI) [21]. The hypothesis that glucose and insulin responses are mainly affected by carbohydrate content was confirmed in our study, where no differences between RED and REG meals were observed. However, others have observed that the quantity of carbohydrates may not be the only factor affecting glucose concentration. In a study with milk, McDonald et al. [22] observed lower glucose response after milk + glucose consumption compared to glucose alone (99 g and 75 g carbohydrates, respectively). In line with our findings, they observed that milk with and without fat did not affect glucose response differently [22]. Thus, their effect on glucose response may arise from other dairy components.

Early epidemiologic studies reported that a high intake of saturated fat was associated with hyperinsulinemia and increased risk of T2D in humans [23–25]. However, intervention studies have not confirmed that increased dietary intake of saturated fatty acids increased the concentration of saturated fatty acids in plasma, the latter being associated with risk of CVD and insulin resistance [3]. More recent findings indicate that the food source may explain the inconsistency in studies, as protective associations between dairy intake and T2D have been found [26,27]. In an intervention study, Itoh et al. showed that a diet high in saturated fat induced higher insulin secretion in Japanese women, despite unchanged

glucose levels, compared with a low-fat control diet [28]. In contrast, we did not observe a rise in insulin caused by a higher saturated fatty acid content in the REG meal compared with the RED meal (saturated fatty acid 14.6 g vs. 4.6 g, respectively) and neither did McDonald et al. [22] observed postprandial differences between a non-fat milk and a full-fat milk.

Dairy proteins have been shown to be important in managing postprandial glucose responses in individuals with and without T2D [29]. The main protein source in cheese is casein and studies investigating effects of casein on insulin and glucose are conflicting [30]. This may be explained by casein consumed in different forms, for example as supplements (hydrolysate, isolate and concentrate) or intact in whole dairy products. Casein is rich in the amino acid proline which induces smaller glucose, but larger insulin responses when added to a glucose-rich meal, compared to glucose alone [31]. Furthermore, mixtures of amino acids are suggested to exert synergistic effects [29]. Gannon et al. [32] provided evidence that dairy food containing ~25 g protein, primarily from low-fat cottage cheese, might be a potent insulin secretagogue when consumed with carbohydrates in participants with diabetes. Our participants had indices of the metabolic syndrome, thus were neither categorised as having diabetes nor being entirely healthy. Although the protein content of the RED meal was higher than the REG meal (30.8 g vs. 24.2 g) and the amount of carbohydrates was similar in the two meals, we did not observe differences in the insuline response between these two cheese meals. However, the potential effect of 6.6 g extra protein may be too small to exert any effect. Still, the effect of normal physiological amounts consumed by a metabolically vulnerable population has yet to be determined.

Most knowledge on hypertriglyceridemia is based on measurements in the fasting state with fasting TG levels of > 1.70 mmol/L being closely related to the development of T2D [33]. In the current study 8 participants had fasting TG > 1.70 mmol/L. One could hypothesize that fasting TG after the 12-week consumption would increase in a dose-dependent matter according to fat intake, however, this was not observed in the main study [11]. Postprandial measurements are clinically relevant, since most individuals in the Western countries spend a considerable amount of non-sleeping time in the postprandial state. TG concentrations measured at 2–4 h after a meal [34] and elevated concentrations that continue into the late postprandial phase [35] have been proposed to predict the risk of developing CVD in humans. Furthermore, postprandial hypertriglyceridemia predicts T2D in rats [36]. Postprandial TG responses to the two cheese meals and the carbohydrate meal differed after 3 and 4 h, reflecting the higher fat intake in the meals (REG: 25.1 g, RED: 16.3 g, CHO: 3.5 g). In fact, participants in the REG group exhibited a prolonged TG response with increasing values even at 4 h. The prolonged TG response after the REG meal is probably due to the higher fat and energy content causing a delay in gastric emptying and prolonged appearance of TG in the blood stream. Whether an increased postprandial TG concentration increases the risk of T2D in humans is unknown and need further investigation. However, it is highly complicated to study these postprandial effects on long-term hard endpoints as many cohorts have not collected the data.

For postprandial FFA response, expected differences between the CHO and cheese meals were observed. A difference between the REG and RED meal was observed at 3 h, but because the glucose and insulin responses for RED and REG were similar this difference in FFA cannot be explained by these and it could be a chance finding.

A recent meal study [37] found that total energy intake (breakfast test meal + ad libitum meal) was reduced when a breakfast meal including a high protein-low fat cheese was consumed, compared to a high protein-high fat cheese. They concluded that protein content, regardless of fat content, increased satiety. In the current study, no effect on subjective appetite sensations was observed. For the comparison of cheeses, one explanation could be that the RED meal, despite the lower energy content, would not reduce satiety (and cause compensatory eating in the next meal) because of the higher protein content, compared to the REG meal. However, we had expected differences between the cheese and CHO meals, so the lack of difference in appetite rating between the meals might be due to insufficient

statistical power [16]. Thus, more studies with appetite sensations as primary outcome are needed to determine the effect of cheese with different fat content on satiety acutely and in the long-term.

To our knowledge, this is the first study to compare the effects of regular-fat cheese with reduced-fat cheese on postprandial changes in insulin and glucose after long-term consumption. The strength of the current study is that it strived to mimic a real-life situation by not matching the breakfast meals for energy and macronutrient composition. We, thereby, studied the metabolic profiles in a situation as close as possible to living on such diets, rather than studying a standardized meal not similar to the three intervention diets. The latter could have posed an interesting research question too, but it was not the purpose of the present study. A limitation of the present study includes the fact that it is exploratory, thus no power calculation was used which increases the risk of false negative results. Additionally, the number of participants was low which limits the possibility of analysing and reporting data disaggregated by sex. Further, the participants were less metabolically vulnerable at week 12 compared to their status at inclusion. Additionally, no standardized meals were provided for the participants in the evening prior to the test day, and that might have affected their fasting values. Furthermore, it was not possible to blind the participants due to the nature of the meals. Finally, it would have been interesting to conduct the meal test not only after the 12-week intervention period, but also at baseline (week 0) to examine if "long-term" intake had influenced the postprandial response. Without the suggested baseline test, it is not possible to separate the effects caused by the 12-week intervention period from the acute effects of the three different test meals.

## 5. Conclusions

In this exploratory sub-study, 12-week consumption of regular-fat cheese by a metabolically vulnerable population did not increase postprandial levels of glucose and insulin compared with consumption of an equal amount of reduced-fat cheese. However, regular-fat cheese resulted in elevated postprandial TG levels. Subjective appetite sensations were not affected by the meals.

**Supplementary Materials:** The following supporting information can be downloaded at: https://www.mdpi.com/article/10.3390/dairy4010004/s1, Table S1: Palatability assessments of the three test meals; Figure S1: Postprandial blood responses of glucose and insulin after intake of regular-fat cheese (REG), reduced-fat cheese (RED) or carbohydrate-rich (CHO) meals (*n* = 37); Figure S2: Postprandial blood responses of triglycerides (TG) and free fatty acids (FFA) after intake of regular-fat cheese (REG), reduced-fat cheese (RED) or carbohydrate-rich (CHO) meals (*n* = 37); Figure S3: Postprandial appetite sensation of satiety (*n* = 37) and hunger (*n* = 36).

**Author Contributions:** Conceptualization, A.R., T.T. and F.R.; methodology, A.R., T.T. and F.R.; software, Dankost, Acqui and R; validation, L.K. and F.R.; formal analysis, F.R., C.R. and L.K.; investigation, F.R. and R.C.J.R.; resources, A.R., T.T. and A.A.; data curation, L.K., F.R. and R.C.J.R.; writing—original draft preparation, F.R.; writing—review and editing, L.K.; visualization, L.K.; supervision, A.R. and T.T.; project administration, F.R.; funding acquisition, A.R., T.T. and A.A. All authors have read and agreed to the published version of the manuscript.

**Funding:** This research was funded 50% by the Danish Dairy Research Foundation, Danish Agriculture and Food Council (Denmark) and 50% by the Dairy Research Institute (United States), the Dairy Farmers of Canada (Canada), Centre National Interprofessionel de l'Economie Laitière (France), Dairy Australia (Australia), and Nederlandse Zuivel Organisatie (The Netherlands).

**Institutional Review Board Statement:** The study was conducted in accordance with the Declaration of Helsinki, and approved by the Ethical Committee of Region H, Denmark (H-4-2013-099, approved 21 November 2014).

**Informed Consent Statement:** Informed consent was obtained from all participants involved in the study.

**Data Availability Statement:** The data presented in this sub-study are available on request from the corresponding author. The data are not publicly available before it is fully anonymised. Pseudo-anonymized data described in the article will be made available before 2025 via a data sharing contract. From 2025 fully anonymized can be transferred or will be made publicly available.

**Acknowledgments:** We thank our kitchen staff Charlotte Kostecki and Karina Graff Rossen for preparing the breakfast meals and our laboratory technicians Aminah Ishaq Palic for collecting and analysing blood samples.

**Conflicts of Interest:** A.A. has received research grants from Arla Foods AMBA, Denmark; The Danish Dairy Research Foundation, Denmark; Global Dairy Platform, USA; the Danish Agriculture and Food Foundation, Denmark, and Arla Food for Health, Denmark. Furthermore, A.A. is Chairman of Scientific Advisory Board, RNCP, Groupe Éthique et Santé, FR (2017-), International Egg Commission/Danske Æg, Expert Group (2020-), Editorial Board Member, Obesity Medicine (2022-), Consultant: Novo Nordisk A/S (2009-), Member of the Board and shareholder in the consultancy company Dentacom Aps, Denmark (2005-), Co-founder and co-owner of the University of Copenhagen spin-out company Mobile Fitness A/S, Denmark (2005-Filed for bankruptcy November 2017), Co-founder, co-owner and Member of the Board of the University of Copenhagen spin-out company Flaxslim ApS, Denmark (2014-), Co-owner of the University of Copenhagen spin-out company Personalized Weight Management Research Consortium ApS (Gluco-Diet.dk) (2017–2020), Recipient of stock options in Gelesis, USA (2018-) and Co-inventor of patents. T.T. has received research grants from Arla Foods AMBA, Denmark; The Danish Dairy Research Foundation, Denmark, and the Dairy Research Institute, USA. A.R. has received research funding from the Dairy Research Institute, USA; The Danish Agriculture and Food Council, Denmark, and Arla Food for Health, Denmark, and A.R. has received consultancy/speaker honoraria from Unilever, Nestlé, and the International Sweeteners Association. L.K. has received grants for research from Arla Foods A/S, Denmark; The Danish Dairy Research Foundation, Denmark, and Arla Food for Health, Denmark. F.R. (employed by this grant), R.C.J.R. and C.R. declare no conflict of interest. The funders were invited to comment on the study design, but the researchers made the final decisions. Thus, the funders had no role in the design of the study; in the collection, analyses, or interpretation of data; in the writing of the manuscript, or in the decision to publish the results.

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
