# Peer review of "Postprandial Glycaemia, Insulinemia, and Lipidemia after 12 Weeks’ Cheese Consumption: An Exploratory Randomized Controlled Human Sub-Study"

_2624-862X, doi:10.3390/dairy4010004_

Round 1

Reviewer 1 Report

Overall evaluation

Kjølbæk et al. take a simple approach to address the metabolic impact of cheese on the human organism by analyzing, in a randomized controlled parallel human intervention sub-study, the postprandial response of blood glucose/insulin and TG/FFA to the ingestion of a bread accompanied with either regular-fat cheese, reduced-fat cheese, or jam. The results delivered by this experiment deliver unspectacular results (because they are expected), namely that (i) regular-fat cheese ingestion results in higher postprandial TG than reduced-fat cheese and jam and (ii) jam ingestion results in higher postprandial glucose and insulin than regular-fat cheese and reduced-fat cheese.

Despite these unspectacular results the Discussion of the manuscript by Kjølbæk et al. provides a sound rationale for the value of the study. This rationale is based on previous reports on dairy products evaluating the role of proteins, fat, carbohydrates, and the fermentation status - as well as their interaction - on the measured postprandial cardiovascular risk factors.

The following points should be clarified/improved:

Title

The title should be reevaluated as, although the cohort is in a real-life situation, the challenge tests are acute tests that are well controlled. It is not clear how a controlled acute trial could mimic a real-life situation.

Introduction

The third paragraph presents the rationale for the current manuscript which is that “None of the matrix studies compared cheese with similar matrix, but different fat content”. Although this rationale is adequate to motivate the present study, as described in the Introduction, the above argument is overstated as this aim has indeed already been published by the authors in 2016 (Raziani et al, Ref 11). Eventually, a postprandial study comparing cheese with different far contents is novel.

Lines 91-92: The Materials and Methods in lines 178-189 provide quite a detailed list of parameters to evaluate appetite sensations. The rationale for measuring appetite sensations during the 4h post-meal is, however, is not provided in the Introduction. The fact that the three test meals (although different in their composition, at least when jam is compared to cheese) did mostly not differ in the sensory evaluation (Table 3 and Figure 4) argues against the utility for conducting such test, unless properly justified. Lines 437-439 in the Discussion state “A recent meal test [40] found that total energy intake (breakfast test meal + ad libitum meal) was reduced when a breakfast meal including a high protein-low fat cheese was consumed, compared to a high protein-high fat cheese”. This information could be provided in the Introduction to justify the sensory analyses.

Materials and Methods

Lines 198-200 state that “All repeated outcomes were analysed as repeated measurements and as 4-hour incremental area under the curve (iAUC) or 4-hour incremental area over the curve (iAOC).” Please precise for parameters, such as glucose whose values run above (first hour) and below (hours 2-4) the fasting value, how iAUC and iAOC take such kinetic patterns into consideration. This explanation should, for example, clarify how the iAUC was calculated for glucose in panels c and d of Figure 2.

Discussion

In lines 379-381 the authors state “In the present sub-study the macronutrient composition was not matched across the three test meals. This was on purpose, since the objective was to evaluate the effects of regular consumption of cheese as a whole food eaten in a real-life situation”. This point is made in the Discussion what indirectly implies, despite the statement of the authors, that it is a limitation of the study. The statement of the authors should thus be brought in the Introduction (with my reserve above on the use of the wording “real-life”) rather than presented as a limitation.

Lines 396-406, discussing insulin, present the concept that amino acids in dairy proteins are insulinotropic but that the contribution of these amino acids within the dairy matrix to circulatory insulin is complex. This section indirectly implies that, although, a higher postprandial insulin response, eventually similar than with jam ingestion (?), could have been expected, this was not the case; hence the complexity mentioned by the authors. The thoughts of the authors on this point should be more clearly presented in this section.

Lines 224-225 state that, out of 37 completers, “12 completers did not have risk factors for metabolic syndrome when the meal test was conducted at week 12”. As the main study aimed at recruiting subjects with at least one risk factor for the metabolic syndrome, in addition to elevated waist circumference, the consequence of having this rather healthy group on the sub-study aims and results should be critically discussed.

In line 416-421 the authors develop the argument that postprandial TGs are clinically relevant as they predict the risk of developing CDV in humans. On the other hand, in lines 423-426, whereas their results show elevated postprandial TGs after intake of regular-fat cheese, the authors drive the discussion towards the non-significant difference in fasting TG levels, instead of discussing the potential clinical consequence of the increased postprandial response. This discussion should be conducted more critically: are the fasting data more relevant risk predictors than the postprandial data? If not, should the postprandial results of the authors be interpreted as an elevated risk for CVD for the consumption of regular-fat cheese compared to reduced-fat cheese? Or is the “real-life situation” described by the authors for the meal challenge to be mitigated with regard to the number of such meals that are normally taken by consumers on a daily/weekly basis?

In the last sentence of the Discussion on lines 460-462, the authors state “Finally, it would have been interesting to conduct the meal test not only after the 12-week intervention period, but also at baseline (week 0) to examine if “long-term” intake had influenced the postprandial response.” This sentence points to a major limitation of the study, which is that the study design contains two factors potentially influencing the results, namely the fact that, in addition to the different meals selected for the postprandial tests, the 12 weeks preceding the postprandial test were also conducted with different diets enriched with the test products used for the postprandial test (lines 138-140). By doing so, the authors introduce 2 factors that could lead to a different postprandial response in glucose, insulin, TG, or fatty acids, namely the challenge itself, but also the preceding 12-w dietary intervention, even the more since there was no washout phase before the challenge test. In disagreement with the proposal of the authors on lines 459-462 to conduct the tests with the three meals also at baseline, more appropriate study designs would have been to either (i) used the same meal challenge to test the impact of the 12-w intervention on the metabolic fitness of the subjects or (ii) have a washout period with the same dietary regimen for all subjects, before testing the three meals postprandially. Using the same meal challenge for the postprandial test would have allowed to identify subtle changes in the postprandial response resulting from a different metabolic status induced by the 12-w dietary regimen (see for example Kardinaal et al. FASEB J. 2015; 29:4600; Fiamoncini et al. FASEB J. 2018; 32:5447). On the other hand, the second study design would have been the most appropriate in regard to the aim of the present study; it would, however, evidently, not have been possible given that this sub-study is derived from a pre-existing study. These limitations should be better presented in the Discussion.

Reviewer 2 Report

The manuscript describes an exploratory sub-study, carried out in a sub-group of people participating in a bigger intervention study with the aim of investigating the effect of cheese with different fat content on postprandial changes in type-2-diabetes risk markers.

Despite the limitations of study (as authors recognise, the fact that it is exploratory, the small number of participants…) I find it interesting because few similar studies have been published.

In addition, the manuscript is very clearly written and easy to read.

Nevertheless, authors should clarify some questions and improve some aspects of the manuscript.

As a general question, in my opinion, the non-cheese carbohydrate-rich (CHO) diet is not a suitable diet as a control. The macronutrient composition is too different compared to the other two diets and the result obtained with it do not help to understand the effects of the fat content of the cheese on the measured parameters. The authors said they try to "imitate a real life situation." I would not say that a  diet with 83% energy in the form of carbohydrates is a frequent diet.

In addition, as a main result they state that "cheese intake with regular compared to reduced fat content did not affect glucose, insulin and appetite, but increased TG and FFA". The increase in TG is expected taking into account the composition of the diets. The increase in FFA is not as predictable, nor is the lack of significant differences in glucose and insulin. For this reason, I have missed in the discussion hypotheses that could explain these results.

More specific comments

M&M

Line 129:  The should describe the population where the participants were selected from, and how the participants  were recruited.

Lines 170-171: The kit they describe fo rTG analysis is not for TG but for cholesterol analysis.

Line 174:  The basis of FFA analysis  should be mentioned.

Results

Line 227: At what point in the study were the data collected? At the beginning of the main study? Just before the test meal?...

Table 2:  All  MetS risk factors and the way they are expressed should be explained in the foot of the table.

Line 241: In Figure 2a it can be seen that the drop in glucose concentration for CHO is steeper after 120 min than the other two. Is not this difference significant?

The footer of figures 2, 3 and 4 includes some sentences that describe the results. In my opinion, they should be removed as the results are explained in the text.

Discussion

In general, authors focus the discussion in comparing their results with other similar studies and discuss if they agree or not. However, in my opinion, the discussion should focus more on hypothesizing the metabolic and physiological reasons for the results.

Line 358: …showed lower glucose and insulin response compared with the CHO meal.  This result is not surprising at all, giving the carbohydrate content of CHO diet.

Lines 354-377: The design and diets in Thurning's study are quite different compared to the present study, making it difficult to draw conclusions. Perhaps more important to note is the fact that there are no similar studies. On the other hand, I find interesting the mention of McDonald et al. work, and I wonder if they give any possible explanation for their results and if the explanation would be applicable to explain the results of the present study.

Lines 377-378:  I do not understand what conclusion the authors want to express with the last sentence of the paragraph

Lines 396-406:  The low protein content of the control diet, compared to the cheese diet, does not allow them to conclude whether it is a specific effect of dairy proteins, which would have been an interesting conclusion. This relates to my general comment that the control diet has not been well designed.

Lines 340-341: Although not significant, maybe due to the small sample size, the fasting TG values seems to differ between the three groups in the current study. They cannot state that if there is no statistical difference, neither the statement in lines 433-436.

Line 441:… despite slight differences in postprandial kinetics.

In results, authors affirm “There were neither time-meal interactions nor main effects of meal on any of the appetite sensation parameters". So, no reason to try to explain differences.

Other minor comments are marked in the revised manuscript

Round 2

Reviewer 1 Report

My comments have been considered and appropriateley addressed

Author Response

Dear Reviewer,

We would like to thank you for the thorough and helpful review of our manuscript. As additional comments have not been raised we consider the review process finalized. 
Merry X-mas.

Kind Regards,

Louise Kjølbæk, corresponding author

Reviewer 2 Report

The authors have adequately responded to most of the questions raised in my review.

Nevertheless, there are a couple of questions that I would like to raise.

In the previous revision, I commented that the non-cheese carbohydrate-rich (CHO) diet is not a suitable diet as a control because its composition is too different from the other two diets. The authors, in their response, correctly justify the inclusion of this diet. But, in any case, I would not call it "control", but just “carbohydrate-rich diet”

On the other hand, the captions of figures 2, 3 and 4 include some sentences that describe the results that, in my opinion, should be removed. The authors disagree and have kept these sentences.

In the section “Instructions for Authors” in the web page of Dairy says “All Figures, Schemes and Tables should have a short explanatory title and caption”.  In addition, in my experience, in most scientific publications, the captions include the information needed to understand the meaning of the figure, but the results are described in the text. For all these reasons, I think that it is up to the editor to decide what the authors should include in the commented figure captions.

A few other minor comments are marked in the revsed manuscript

Author Response

Dear Reviewer,

We would like to thank you for the second thorough review of our manuscript.
We have addressed all comments in the point-by-point reply below and we have used the “track changes”-function in the revised manuscript.

Kind Regards,
Louise Kjølbæk, corresponding author

Comments

In the previous revision, I commented that the non-cheese carbohydrate-rich (CHO) diet is not a suitable diet as a control because its composition is too different from the other two diets. The authors, in their response, correctly justify the inclusion of this diet. But, in any case, I would not call it "control", but just “carbohydrate-rich diet”

Reply: We have changed “control” to “meal”.

On the other hand, the captions of figures 2, 3 and 4 include some sentences that describe the results that, in my opinion, should be removed. The authors disagree and have kept these sentences.
In the section “Instructions for Authors” in the web page of Dairy says “All Figures, Schemes and Tables should have a short explanatory title and caption”.  In addition, in my experience, in most scientific publications, the captions include the information needed to understand the meaning of the figure, but the results are described in the text. For all these reasons, I think that it is up to the editor to decide what the authors should include in the commented figure captions.

Reply: We have asked our contact person Regina Li who is Section Managing Editor for advice. We received this reply:

Dear Mrs. Kjølbæk,
Thanks for your email. The caption in the figures can be kept without changing, and you can modify the manuscript according to other comments of the reviewer. We look forward to receiving the revised manuscript.
Kind regards,
Regina

Based on this reply we have not made any changes.

A few other minor comments are marked in the revised manuscript.

Reply: Thank you so much.